# *Few-Class Arena*: A Benchmark for Efficient Selection of Vision Models and Dataset Difficulty Measurement

## Abstract

A wide variety of benchmark datasets with many classes (80-1000) have been created to assist Computer Vision architectural evolution. An increasing number of vision models are evaluated with these many-class datasets. However, real-world applications often involve substantially fewer classes of interest (2-10). This gap between many and few classes makes it difficult to predict performance of the few-class applications using models trained on the available many-class datasets. To date, little has been offered to evaluate models in this *Few-Class Regime*. We propose *Few-Class Arena* (*FCA*), as a unified benchmark with focus on testing efficient image classification models for few classes. We conduct a systematic evaluation of the ResNet family trained on ImageNet subsets from 2 to 1000 classes, and test a wide spectrum of Convolutional Neural Networks and Transformer architectures over ten datasets by using our newly proposed *FCA* tool. Furthermore, to aid an up-front assessment of dataset difficulty and a more efficient selection of models, we incorporate a difficulty measure as a function of class similarity. *FCA* offers a new tool for efficient machine learning in the *Few-Class Regime*, with goals ranging from a new efficient class similarity proposal, to lightweight model architecture design, to a new scaling law. *FCA* is user-friendly and can be easily extended to new models and datasets, facilitating future research work. Our benchmark is available at `https://github.com/fewclassarena/fca`.

## 1 Introduction

The de-facto benchmarks for evaluating efficient vision models are large scale with many classes (e.g. 1000 in ImageNet [1], 80 in COCO [2], etc.). Such benchmarks have expedited the advance of vision neural networks toward efficiency [3, 4, 5, 6, 7, 8, 9, 10] with the hope of reducing the financial and environmental cost of vision models [11, 12]. More efficient computation is facilitated by using quantization [13, 14, 15], pruning [16, 17, 18, 19], and data saliency [20]. Despite efficiency improvements such as these, many-class datasets are still the standard of model evaluation.

Real-world applications, however, typically comprise only a few number of classes (e.g, less than 10) [21, 22, 23] which we termed *Few-Class Regime*. To deploy a vision model pre-trained on large datasets in a specific environment, it requires the re-evaluation of published models or even retraining to find an optimal model in an expensive architectural search space [24].

One major finding is that, apart from scaling down model and architectural design for efficiency, dataset difficulty also plays a vital role in model selection [25] (described in Section 4.3).

Submitted to the 38th Conference on Neural Information Processing Systems (NeurIPS 2024) Track on Datasets and Benchmarks. Do not distribute.

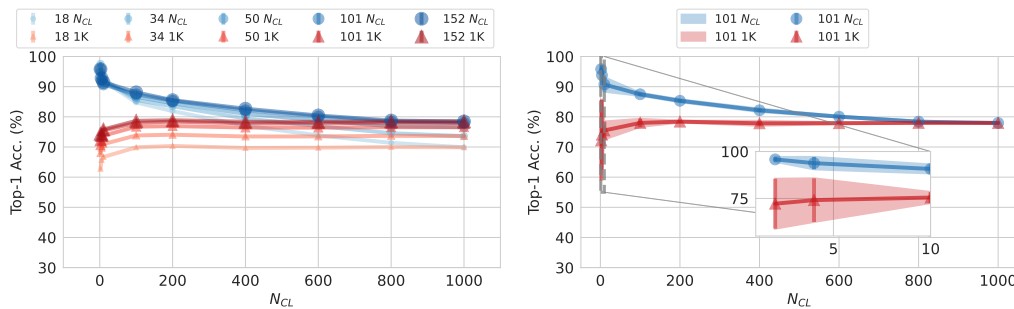

(a) Accuracies for sub-models (blue) and full models (red).

(b) Zoomed window shows accuracy values and range for full and sub-models in the few-class range.

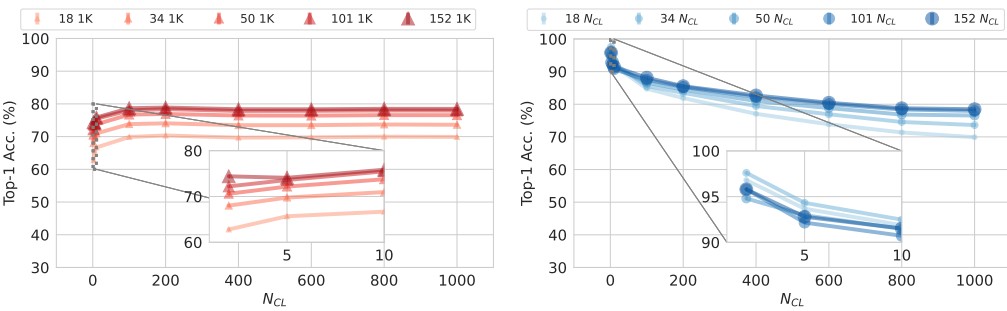

(c) Zoomed window shows (c.1) drop of accuracy as $N_{CL}$ decreases, (c.2) accuracy scales with model size for full models in the few-class range.

(d) Zoomed window shows (d.1) rising accuracy as $N_{CL}$ decreases, (d.2) accuracy does not scale with model size for sub-models in the few-class range.

Figure 1: Top-1 accuracies of various scales of ResNet, whose model sizes are shown in the legend, and whose plots vary from dark to light by decreasing size. Plots range along number of classes $N_{Cl}$ from the full ImageNet size (1000) down to the *Few-Class Regime*. Each model is tested on 5 subsets whose $N_{Cl}$ classes are randomly sampled from the original 1000 classes. (a) Plots for sub-models trained on subsets of classes (blue) and full models trained on all 1000 classes (red). (b) Zoomed window shows the standard deviation of subset's accuracies is much smaller than for the full model. (c.1) Full model accuracies drop when $N_{CL}$ decreases. (c.2) Full model accuracies increase as model scales up in the *Few-Class Regime*. (d.1) Sub-model accuracies grow as $N_{CL}$ decreases. (d.2) Sub-model accuracies do not increase when model scales up in the *Few-Class Regime*.

Figure 1 summarizes several key findings under the *Few-Class Regime*. On the left graph in red are accuracy results for a range of number of classes $N_{CL}$ for what we call the "full model", that is ResNet models pre-trained on the full 1000 classes of ImageNet (generally available from many websites). On the right are accuracy results for what we call "sub-models", each of which is trained and tested on the same $N_{CL}$, where this number of classes is sampled from the full dataset down to the *Few-Class Regime*. Findings include the following. (a) Sub-models attain higher upper-bound accuracy than full models. (b) The range of accuracy widens for full models at few-classes, which increases the uncertainty of a practitioner selecting a model for few classes. In contrast, sub-models narrow the range. (c) Full models follow the scaling law [26] in the dimension of model size - larger models (darker red) have higher accuracy from many to few classes. (4) Surprisingly, the scaling law is violated for sub-models in the *Few-Class Regime* (see the zoomed-in subplot) where larger models (darker blue) do not necessarily perform better than smaller ones (lighter blue). From these plots, our key insight is that, instead of using full models, researchers and practitioners in the *Few-Class Regime* should use sub-models for selection of more efficient models.

However, obtaining sub-models involves computationally expensive training and testing cycles since they need to be converged on each of the few-class subsets. By carefully studying and comparing the experiment and evaluation setup of these works in the literature, we observe that, how models scale down to *Few-Class Regime* is rarely studied. The lack of comprehensive benchmarks for *few-class* research impedes both researchers and practitioners from quickly finding models that are the most

efficient for their dataset size. To fill this need, we propose a new benchmark, *Few-Class Arena* (*FCA*), with the goal of benchmarking vision models under few-class scenarios. To our best knowledge, *FCA* is the first benchmark for such a purpose.

We formally define *Few-Class Regime* as a scenario where the dataset has a limited number of classes. Real-world applications often comprise only a few number of classes (e.g. $N_{CL} < 10$ or $10\%$ classes of a dataset). Consequently, *Few-Class Arena* refers to a benchmark to conduct research experiments to compare models in the *Few-Class Regime*. This paper focuses on the image classification task, although *Few-Class Regime* can generalize to object detection and other visual tasks.

**Statement of Contributions.** Four contributions are listed below:

- To be best of our knowledge, we are the first to explore the problems in the *Few-Class Regime* and develop a benchmark tool *Few-Class Arena* (*FCA*) to facilitate scientific research, analysis, and discovery for this range of classes.
- We introduce a scalable few-class data loading approach to automatically load images and labels in the *Few-Class Regime* from the full dataset, avoiding the need to duplicate data points for every additional few-class subset.
- We incorporate dataset similarity as an inverse difficulty measurement in *Few-Class Arena* and propose a novel Silhouette-based similarity score named *SimSS*. By leveraging the visual feature extraction power of CLIP and DINOv2, we show that *SimSS* is highly correlated with ResNet performance in the *Few-Class Regime* with Pearson coefficient scores $\geq 0.88$.
- We conduct extensive experiments that comprise ten models on ten datasets and 2-1000 numbers of classes on ImageNet, totalling 1591 training and testing runs. In-depth analyses on this large body of testing reveal new insights in the *Few-Class Regime*.

## 2 Related Work

**Visual Datasets and Benchmarks.** To advance deep neural network research, a wealth of large-scale many-class datasets has been developed for benchmarking visual neural networks over a variety of tasks. Typical examples [1] include 1000 classes in ImageNet [1] for image classification, and 80 object categories in COCO [2] for object detection. Previous benchmarks also extend vision to multimodal research such as image-text [27, 28, 29, 30]. While prior works often scale up the number of object categories for general purpose comparison, studies [31, 32] raise a concern on whether models trained on datasets with such a large number of classes (e.g. ImageNet) can be reliably transferred to real world applications often with far fewer classes. A close work to ours is vision backbone comparison [33] whose focus is on model architectures. Our perspective differs in a focus on cases with fewer number of classes, which often better aligns with real-world scenarios.

**Dataset Difficulty Measurement.** Research has shown the existence of inherent dataset difficulty [32] for classification and other analytic tasks. Efficient measurement methods are proposed to characterize dataset difficulty using Silhouette Score [34], K-means Fréchet inception distance [35, 36, 37], and Probe nets [25]. Prior studies have proposed image quality metrics using statistical heuristics, including peak signal-to-noise ratio (PSNR) [38], structural similarity (SSIM) Index [39], and visual information fidelity VIF [40]. A neuroscience-based image difficulty metric [32] is defined as the minimum viewing time related to object solution time (OST) [41]. Another type of difficulty measure method consists of additional procedures such as c-score [42] , prediction depth [43] , and adversarial robustness [44] . Our work aligns with the line of research [45, 46, 47] involving similarity-based difficulty measurements: similar images are harder to distinguish from each other while dissimilar images are easier. Previous studies are mainly in the image retrieval context [48, 49, 50]. Similarity score is used in [51] with the limitation that a model serving similarity measurement has to be trained for one dataset. We push beyond this limit by leveraging large vision models that learn general visual features using CLIP [52] and DINOv2 [53]. The study [32] shows that CLIP generalizes well to both easy and hard images, making it a good candidate for measuring

---

[1]A detailed list of many-class datasets used in this paper can be found in the Appendix.

image difficulty. Supported by the evidence that better classifiers can act as better perceptual feature extractors [54] , in later sections we show how CLIP and DINOv2 will be used as our similarity base function.

Despite the innovation of difficulty measure algorithms on many-class datasets, little attention has been paid to leveraging these methods in the *Few-Class Regime*. We show that, as the number of classes decreases, sub-dataset difficulty in the *Few-Class Regime* plays a more critical role in efficient model selection. To summarize, unlike previous work on many-class benchmarks and difficulty measurements, our work takes few-class and similarity-based dataset difficulty into consideration, and in doing so we believe the work pioneers the development of visual benchmark dedicated to research in the *Few-Class Regime*.

## 3 Few-Class Arena (FCA)

We introduce the *Few-Class Arena* (*FCA*) benchmark in this section. In practice, we have integrated *FCA* into the MMPreTrain framework [55], implemented in Python3 and Pytorch[2].

### 3.1 Goals

**1. Generality.** All vision models and existing datasets for classification should be compatible in this framework. In addition, users can extend to custom models and datasets for their needs.

**2. Efficiency.** The benchmark should be time- and space-efficient for users. The experimental setup for the few-class benchmark should be easily specified by a few hyper-parameters (e.g. number of classes). Since the few-class regime usually includes sub-datasets extracted from the full dataset, the benchmark should be able to locate those sub-datasets without generating redundant duplicates for reasons of storage efficiency. For time-efficiency, it should conduct training and testing automatically through use of user-specified configuration files, without users' manual execution.

**3. Large-Scale Benchmark.** The tool should allow for large-scale benchmarking, including training and testing of different vision models on various datasets when the number of classes varies.

### 3.2 Few-Class Dataset Preparation

*Few-Class Arena* provides an easy way to prepare datasets in the *Few-Class Regime*. By leveraging the MMPreTrain framework, users only need to specify the parameters of few-class subsets in the configuration files, which includes the list of models, datasets, number of classes ($N_{CL}$), and the number of seeds ($N_S$). *Few-Class Arena* generates the specific model and dataset configuration files for each subset, where subset classes are randomly extracted from the full set of classes, as specified by the seed number. Note that only one copy of the full, original dataset is maintained during the whole benchmarking life cycle because few-class subsets are created through the lightweight configurations, thus maximizing storage efficiency. We refer readers to the Appendix and the publicly released link for detailed implementations and use instructions.

### 3.3 Many-Class Full Dataset Trained Benchmark

We conducted large-scale experiments spanning ten popular vision models (including CNN and ViT architectures) and ten common datasets [3]. Except for ImageNet1K, where pre-trained model weights are available, we train models in other datasets from scratch. While different models'

---

[2]Code is available at `https://github.com/fewclassarena/fca`, including detailed documentation and long-term plans of maintenance.

[3]Models include: ResNet50 (RN50), VGG16, ConvNeXt V2 (CNv2), Inception V3 (INCv3), EfficientNet V2 (EFv2), ShuffleNet V2 (SNv2), MobileNet V3 (MNv2), Vision Transformer base (ViTb), Swin Transformer V2 base (SWv2b) and MobileViT small (MViTs). Datasets include CalTech101 (CT101), CalTech256 (CT256), CIFAR100 (CF100), CUB200 (CB200), Food101 (FD101), GTSRB43, (GT43), ImageNet1K (IN1K), Indoor67 (ID67), Quickdraw345 (QD345) and Textures47 (TT47).

training procedures may incur various levels of complexity (particularly in our case for MobileNet V3 and Swin Transformer V2 base), we have endeavored to minimize changes in the existing training pipelines from MMPreTrain. The rationale is that if a model exhibits challenges in adapting it to a dataset, then it is often not a helpful choice for a practitioner to select for deployment.

Results are summarized in Table 1. We make several key observations: (1) models in different datasets (in rows) yield highly variable levels of performance by Top-1 accuracy; (2) no single best model (bold, in columns) exists across all datasets; and (3) model rankings vary across various datasets.

The first two observations are consistent with the findings in [25, 31]. For (1), it suggests there exists underlying dataset-specific difficulty. To capture this characteristic, we adopt the reference dataset classification difficulty number (DCN) [25] to refer to the empirically highest accuracy achieved in a dataset from a finite number of models shown in Table 1 and Figure 2 (a). For observation (3), we can examine the rankings among the ten models of ResNet50 and EfficientNet V2 in Figure 2 (b). ResNet50's ranking varies dramatically for the different datasets, for instance ranking 7th on ImageNet1K and 1st on Quickdraw345. This ranking variability is also observed in other models (see all models in the Appendix). However, a common practice is to benchmark models – even for efficiency – on large datasets, especially ImageNet1K. The varied dataset rankings in our experiments expose the limitations of such a practice, further supporting our new benchmark paradigm, especially in the *Few-Class Regime*. In later sections, we leverage DCN and image similarity for further analysis.

| Dataset | RN50 [56] | VGG16 [57] | CNv2 [58] | INCv3 [59] | EFv2 [4] | SNv2 [9] | MNv3 [7] | ViTb [60] | SWv2b [61] | MViTs [10] | DCN [25] |
|---|---|---|---|---|---|---|---|---|---|---|---|
| GT43 [62] | 99.85 | 96.60 | 99.83 | 99.78 | 99.86 | **99.87** | 5.98 | 99.31 | 99.78 | 99.69 | 99.87 |
| CF100 [63] | 74.56 | 71.12 | **85.89** | 75.97 | 77.05 | 77.89 | 1.00 | 32.65 | 78.49 | 76.51 | 85.89 |
| IN1K [1] | 76.55 | 71.62 | 84.87 | 77.57 | **85.01** | 69.55 | 67.66 | 82.37 | 84.6 | 78.25 | 85.01 |
| FD101 [64] | 83.76 | 75.82 | 63.80 | 83.96 | 80.82 | 79.36 | 0.99 | 52.21 | **84.30** | 82.23 | 84.30 |
| CT101 [65] | 77.70 | 74.99 | 77.52 | 77.52 | 77.82 | **84.13** | 76.58 | 59.59 | 78.82 | 80.06 | 84.13 |
| CT256 [66] | 65.07 | 59.08 | **73.57** | 66.09 | 62.80 | 68.13 | 22.63 | 44.23 | 67.28 | 65.80 | 73.57 |
| QD345 [67] | **69.14** | 19.86 | 62.86 | 68.25 | 68.81 | 67.32 | 0.72 | 19.67 | 66.54 | 68.76 | 69.14 |
| CB200 [68] | 45.86 | 21.26 | 27.61 | 45.58 | 44.48 | 53.95 | 47.22 | 23.73 | 54.52 | **58.46** | 58.46 |
| ID67 [69] | 53.75 | 26.01 | 33.21 | 45.95 | 43.85 | **54.72** | 49.10 | 30.51 | 48.58 | 54.05 | 54.72 |
| TT47 [70] | 30.43 | 12.55 | 6.49 | 14.20 | 21.17 | **43.83** | 2.18 | 31.38 | 33.94 | 24.41 | 43.83 |

Table 1: Top-1 accuracy across ten models in ten datasets. Models are trained and tested on full datasets with their original number of classes (e.g. 1K from ImageNet1K); this is denoted in the last few digits of the abbreviation of the dataset name. The best score is highlighted in bold while the second best is underlined for each dataset.

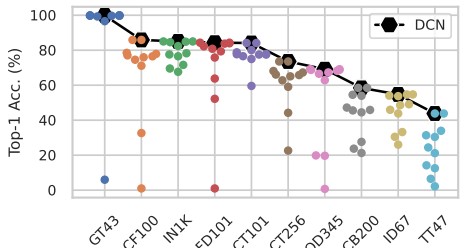
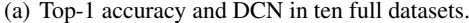
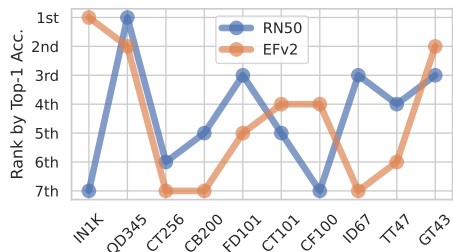

(a) Top-1 accuracy and DCN in ten full datasets.

(b) Ranking of ResNet50 (RN50) and Efficient-Net V2 (EFv2) across 10 datasets by Top-1 acc.

Figure 2: Many-Class Full Dataset Benchmark.

In the next subsections, we introduce three new types of benchmarks: (1) Few-Class, Full Dataset Trained Benchmark (FC-Full), which benchmarks vision models trained on the full dataset with the original number of classes; (2) Few-Class, Subset Trained Benchmark (FC-Sub), which benchmarks vision models trained on subsets of a fewer number of classes than the full dataset, and (3) Few-Class Similarity Benchmark (FC-Sim), which benchmarks image similarity methods and their correlation with model performance.

### 3.4 Few-Class Full Dataset Trained Benchmark (FC-Full)

Traditionally, a large number of models are trained and compared on many-class datasets. However, results for such benchmarks are not directly useful to the *Few-Class Regime* and many real-world scenarios. Therefore, we introduce the Few-Class Full Dataset Trained Benchmark (FC-Full), with the objective of effortlessly conducting large-scale experiments and analyses in the *Few-Class Regime*.

The procedure of FC-Full consists of two main stages. In the first stage, users select the models and datasets upon which they would like to conduct experiments. They can choose to download pre-trained model weights, which are usually available on popular model hubs (PyTorch Hub [71], TensorFlow Hub [72], Hugging Face [73], MMPreTrain [55] etc.). In case of no pre-trained weights available from public websites, users can resort to the option of training from scratch. To that end, our tool is designed and implemented to generate bash scripts for easily configurable and modifiable training through the use of configuration files.

In the second stage, users conduct benchmarking in the *Few-Class Regime*. By specifying the list of classes, *Few-Class Arena* automatically loads pre-trained weights of the chosen models and evaluates performance of the models on the selected datasets. Note that this process is accomplished through configuration files created by the user's specifications, thus enabling hundreds of experiments to be launched by a single command. This dramatically reduces human effort that would otherwise be expended to run these experiments without *Few-Class Arena*.

### 3.5 Few-Class Subset Trained Benchmark (FC-Sub)

Our study in Figure 1 (red lines) reveals the limits of existing pre-trained models in the *Few-Class Regime*. To facilitate further research and analyze the upper bound performance in the *Few-Class Regime*, we introduce the Few-Class Subset Trained Benchmark (FC-Sub).

FC-Sub follows a similar procedure to FC-Full, except that, when evaluating a model in a subset with a specific number of classes, that model should have been trained on that same subset. Specifically, in Stage One (described for FC-Full), users specify models, datasets and the list of number of classes in configuration files. Then *Few-Class Arena* generates bash scripts for model training on each subset. In Stage two, *Few-Class Arena* tests each model in the same subset that it was trained on.

### 3.6 Few-Class Similarity Benchmark (FC-Sim)

One objective of our tool is to provide the Similarity Benchmark as a platform for researchers to design custom similarity scores for efficient comparison of models and datasets.

The intrinsic image difficulty of a dataset affects a model's classification performance (and human) [74, 75, 32]. We show – as is intuitive – that the more similar two images are, the more difficult it is for a vision classifier to make a correct prediction. This suggests that the level of similarity of images in a dataset can be used as a proxy for a dataset difficulty measure. In this section, we first adopt and provide the basic formulation of similarity, the baseline of a similarity metric. Then we propose a Similarity-Based Silhouette Score to capture the characteristic of image similarity in a dataset.

We first adopt the basic similarity formulation from [51]. **Intra-Class Similarity** $S_\alpha^{(C)}$ is defined as a scalar describing the similarity of images within a class by taking the average of all the distinct class pairs in $C$, while **Inter-Class Similarity** denotes a scalar describing the similarity among images in two different classes $C_1$ and $C_2$. For a dataset $D$, these are defined as the mean of their similarity scores over all classes, respectively:

$$S_\alpha^{(D)} = \frac{1}{|L|} \sum_{l \in L} S_\alpha^{(C_l)} = \frac{1}{|L| \times |P^{(C_l)}|} \sum_{l \in L} \sum_{i,j \in C_l;\ i \neq j} \cos(\mathbf{Z}_i, \mathbf{Z}_j), \tag{1}$$

$$S_\beta^{(D)} = \frac{1}{|P^{(D)}|} \sum_{a,b \in L; a \neq b} S_\beta^{(C_a,C_b)} = \frac{1}{|P^{(D)}| \times |P^{(C_1,C_2)}|} \sum_{a,b \in L;\ a \neq b} \sum_{i \in C_1, j \in C_2} \cos(\mathbf{Z}_i, \mathbf{Z}_j). \tag{2}$$

where $|L|$ is the number of classes in a dataset, $Z_i$ is the visual feature of an image $i$, $|P^{(C)}|$ is the total number of distinct image pairs in class $C$, $|P^{(D)}|$ is the total number of distinct class pairs, and $|P^{(C_1,C_2)}|$ is the total number of distinct image pairs excluding same-class pairs.

Averaging these similarities provides a single scalar score at the class or dataset level. However, this simplicity neglects other cluster-related information that can better reveal the underlying dataset difficulty property of a dataset. In particular, the **(1) tightness of a class cluster** and **(2) distance to other classes** of class clusters, are features that characterize the inherent class difficulty, but are not captured by $S_\alpha$ or $S_\beta$ alone.

To compensate the aforementioned drawback, we adopt the Silhouette Score (SS) [34, 76]: $SS(i) = \frac{b(i)-a(i)}{max(a(i),b(i))}$, where $SS(i)$ is the Silhouette Score of the data point $i$, $a(i)$ is the average dissimilarity between $i$ and other instances in the same class, and $b(i)$ is the average dissimilarity between $i$ and other data points in the closest different class.

Observe that the above Intra-Class Similarity $S_\alpha^{(C)}$ already represents the tightness of the class $(C)$, therefore $a(i)$ can be replaced with the inverse of Intra-Class Similarity $a(i) = -S_\alpha(i)$. For the second term $b(i)$, we adopt the previously defined Inter-Class Similarity $S_\beta^{(C_1,C_2)}$ and introduce a new similarity score as **Nearest Inter-Class Similarity** $S'_\beta{}^{(C)}$, which is a scalar describing the similarity among instances between class $C$ and the closest class of each instance in $C$. The dataset-level Nearest Inter-Class Similarity $S'_\beta{}^{(D)}$ is expressed as:

$$S'_\beta{}^{(D)} = \frac{1}{|L|} \sum_{l \in L} S'_\beta{}^{(C_l,\hat{C}_l)} = \frac{1}{|L| \times |P^{(C_l,\hat{C}_l)}|} \sum_{l \in L} \sum_{i \in C_l, j \in \hat{C}_l} \cos(\mathbf{Z}_i, \mathbf{Z}_j). \qquad (3)$$

where $\hat{C}$ is the set of the nearest class to $C$ ($\hat{C} \neq C$). To summarize, we introduce our novel **Similarity-Based Silhouette Score** $SimSS$[4]:

$$SimSS^{(D)} = \frac{1}{|L| \times |C_l|} \sum_{i \in C_l} \frac{S_\alpha(i) - S'_\beta(i)}{max(S_\alpha(i), S'_\beta(i))}. \qquad (4)$$

## 4    Experimental Results

### 4.1    Results on FC-Full

In this section, we present the results of FC-Full. A model trained on the dataset with its original number of classes (e.g. 1000 in ImageNet1K) is referred to as a *full-class model*. These experiments are designed to understand how full-class model performance changes when the number of classes $N_{Cl}$ decreases from many to few classes. We analyze the results of DCN-Full, shown in Figure 3 (details of all models are presented in the Appendix), and we make two key observations when $N_{Cl}$ reduces to the *Few-Class Regime* (from right to left). (1) The best performing models do not always increase its accuracy for fewer classes, as shown by the solid red lines that represent the average of DCN for each $N_{Cl}$. (2) The variance, depicted by the light red areas, of the best models broaden dramatically for low $N_{Cl}$, especially for $N_{Cl} < 10$.

Both observations support evidence of the limitations of using the common many-class benchmark for application model selection in the *Few-Class Regime*, since it is not consistent between datasets that a model can be made smaller with higher accuracy. Furthermore, the large variance in accuracy means that prediction of performance for few classes is unreliable for this approach.

### 4.2    Results on FC-Sub

In this section, we show how using *Few-Class Arena* can help reveal more insights in the *Few-Class Regime* to mitigate the issues of Section 4.1.

---

[4]The extended derivation is detailed in the Appendix.

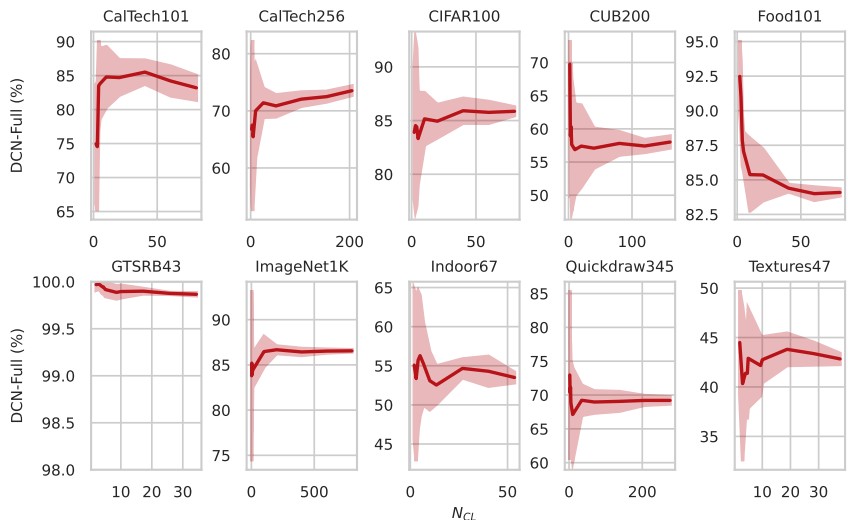

Figure 3: DCN-Full by Top-1 Accuracy (%). $N_{Cl}$ ranges from many to 2.

FC-Sub results are displayed in Figure 4. Recall that a *sub-class* model is a model trained on a subset of the dataset where $N_{Cl}$ is smaller than the original number of classes in the full dataset. Observe that in the *Few-Class Regime* (when $N_{Cl}$ decreases from 4 to 2) that: (1) DCN increases as shown by the solid blue lines, and (2) variance reduces as displayed by the light blue areas.

The preceding observation for FC-Full 4.1 seems to contradict the common belief that, the fewer the classes, the higher is the accuracy that a model can achieve. Conversely, the FC-Sub results do align with this belief. We argue that a full-class model needs to accommodate many parameters to learn features that will enable high performance across all classes in a many-class, full dataset. With the same parameters, however, a sub-class model can adapt to finer and more discriminative features that improve its performance when the number of target classes are much smaller.

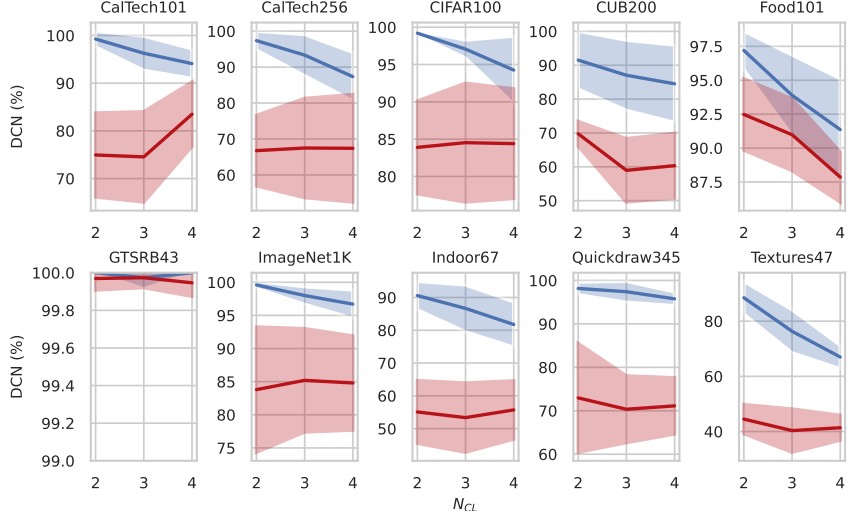

Figure 4: DCN-Sub (red) and DCN-Full (blue) by Top-1 Accuracy (%). $N_{CL}$ ranges from 2 to 4.

## 4.3 Results on FC-Sim

In this section, we analyze the use of SimSS (Equation 4) as proxy for few-class dataset difficulty. Experiments are conducted on ImageNet1K using the ResNet family for the lower $N_{CL} \leq 10\%$ range of the original 1000 classes, $N_{CL} \in \{2, 3, 4, 5, 10, 100\}$, and the results are shown in Figure 5. Each datapoint of DCN-Full (diamond in red) or DCN-Sub (square in blue) represents an experiment in a

subset of a specific $N_{CL}$, where classes are sampled from the full dataset. For reproducible results, we use seed numbers from 0 to 4 to generate 5 subsets for one $N_{CL}$ by default. A similarity base function ($sim()$) is defined as the atomic function that takes a pair of images as input and outputs a scalar that represents their image similarity.

In our experiments, we leverage the general visual feature extraction ability of CLIP (image + text) [52] and DINOv2 (image) [53] by self-supervised learning. Specifically, a pair of images are fed into its latent space from which the the cosine score is calculated and normalized to 0 to 1. Note that we only use the Image Encoder in CLIP.

**Comparing Accuracy and Similarity** To evaluate SimSS, we compute the Pearson correlation coefficient (PCC) ($r$) between model accuracy and SimSS. Results in Figure 5 (a) (b) show that SimSS is poorly correlated with DCN-Full ($r = 0.18$ and $r = 0.26$ for CLIP and DINOv2) due to the large variance shown in Section 4.1. In contrast, SimSS is highly correlated with DCN-Sub (shown in blue squares), with $r = 0.90$ and $r = 0.88$ using CLIP (dashed) and DINOv2 (solid), respectively. The high PCC [77, 78] demonstrates that SimSS is a reliable metric to estimate few-class dataset difficulty, and this can help predict the empirical upper-bound accuracy of a model in the *Few-Class Regime*. Comparison between SimSS and all models can be found in the Appendix. Such a high correlation suggests this offers a reliable scaling relationship to estimate model accuracy by similarity for other values of $N_{CL}$ without an exhaustive search. Due to the dataset specificity of the dataset difficulty property, this score is computed once and used for all times the same dataset is used. We have made available difficulty scores for many datasets at the *Few-Class Arena* site.

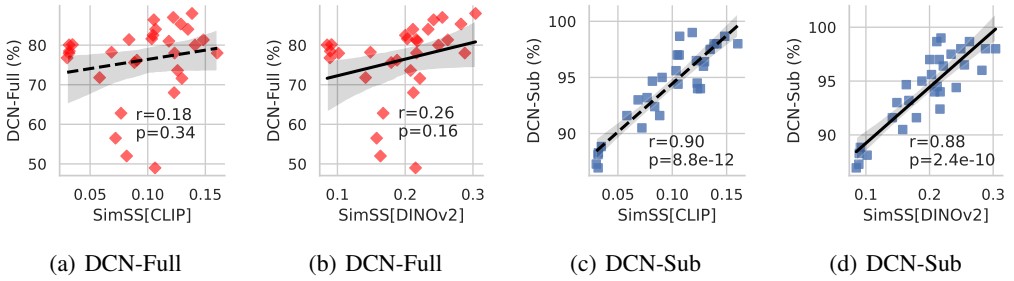

|(a) DCN-Full|(b) DCN-Full|(c) DCN-Sub|(d) DCN-Sub|

Figure 5: Pearson correlation coefficient ($r$) between DCN and SimSS when $N_{Cl} \in \{2, 3, 4, 5, 10, 100\}$. DCN-Sub (blue squares) is more highly correlated than DCN-Full (red diamonds) with SimSS using both similarity base functions of CLIP (dashed line) and DINOv2 (solid line) with $r \geq 0.88$.

## 5 Conclusion

We have proposed *Few-Class Arena* and a dataset difficulty measurement, which together form a benchmark tool to compare and select efficient models in the *Few-Class Regime*. Extensive experiments and analyses over 1500 experiments with 10 models on 10 datasets have helped identify new behavior that is specific to the *Few-Class Regime* as compared to for many-classes. One finding reveals a new $n_{Cl}$-scaling law whereby dataset difficulty must be taken into consideration for accuracy prediction. Such a benchmark will be valuable to the community by providing both researchers and practitioners with a unified framework for future research and real applications.

**Limitations and Future Work.** We note that the convergence of sub-models is contingent on various factors in a training scheduler, such as learning rate. A careful tuning of training procedure may increase a model's performance, but it shouldn't change the classification difficulty number drastically since this represents a dataset's intrinsic difficulty property. The current difficulty benchmark supports image similarity while in the future it can be expanded to other difficulty measurements [25]. As CLIP and DINOv2 are trained toward general visual features, it is unclear if they will be appropriate for other types of images such as sketches without textures in Quickdraw [67] . For this reason, a universal similarity foundation model would be appealing that applies to any image type. In summary, *Few-Class Arena* identifies a promising new path to achieve efficiencies that are focused on the important and practical *Few-Class Regime*, establishing this as a baseline for future work.

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
