# OpenReview forum: "Few-Class Arena: A Benchmark for Efficient Vision Model Selection and Dataset Difficulty"
_NeurIPS.cc/2024/Datasets_and_Benchmarks_Track — Submitted to NeurIPS 2024 Track Datasets and Benchmarks_

### Official Review · Reviewer_zZVd · 2024-07-08
**Few-class benchmark for vision model selection and difficulty evaluation**

**Rating:** 5
**Confidence:** 4

**Review:**

The paper is generally well-structured and clearly written, but the significance is questionable. While addressing the gap in benchmarking efficient image classification models for few-class scenarios is meaningful, the limitations in the flawed interpretations, the lack of comprehensive comparisons with SOTA, and the absence of ablation studies and sensitivity analysis reduce the potential impact of the work.

Clarity:
It is unclear to me how the authors choose which few classes to choose to evaluate model performance.

Originality:
The approach of the work basically brought us back to CIFAR10. This is what you get with a smaller data distribution. The dataset difficulty part is also expected with prior literature. The evaluation metric for difficulty based on class similarity is interesting, but that is based on the feature extraction power of large pre-trained models (CLIP and DINOv2) with many classes. It has to be made clear why do we need a few-class arena and a systematic way to define which classes.

Significance:
Model generalization is an important character of model performance. Evaluation on many classes is an advantage which can evaluate the generalizability to a broader data distribution. Even if the performance is originally not adapted to specific classes, with fine-tuning (many ways of training), a well-trained and generalizable model can quickly adapt to specific tasks. Thus the significance of this work is limited.

Further research is needed to address these concerns and establish the validity of the proposed benchmark and similarity measure.

**Strengths:**

The strengths of this paper include:
1.	Few-class and unified benchmark: Systematically explore the few-class regime (2-10 classes), which is highly relevant to real-world applications but largely overlooked in previous research focused on many-class datasets (80-1000 classes). Few-Class Arena (FCA) provides a unified benchmark that enables systematic evaluation of a diverse range of models (CNNs and Transformers) on multiple datasets across a wide spectrum of class sizes.
2.	Dataset difficulty measure: Proposed a Silhouette-based similarity score (SimSS), which leverages state-of-the-art CLIP and DINOv2 models for assessing dataset difficulty in the few-class regime.

**Additional Feedback:**

For future research, the reviewer would suggest the authors conduct cross-domain evaluation: Investigating how models trained on Few-Class Arena perform on other few-class datasets from different domains (e.g., medical imaging, satellite imagery) could demonstrate the generalizability and robustness of the benchmark.

**Clarity:**

The paper is generally well-written and clear, with a concise abstract and clearly defined goals. While the results and analyses are presented in an organized manner, further elaborating on the uncertainty and sensitivity aspects would strengthen the discussion. Besides, minor typographical errors are present; for instance, the description of Figure 1, line 42, incorrectly indicates “(4)” where the context suggests the correct notation should be "(d)".

**Correctness:**

The research appears to be constructed in a sound way. The evaluation methods and experimental design seem appropriate, with the authors conducting experiments spanning ten models and ten datasets, totaling 1,591 training and testing runs. However, the significance of claims are not fully justified.

**Documentation:**

The authors provide a good amount of documentation on data collection and benchmark organization. The provided code repository, along with the details on dataset preparation and the use of the MMPreTrain framework, offers a solid foundation for reproducibility.

**Ethics:**

Based on the information provided in the paper, there do not appear to be any major ethical concerns.

**Limitations:**

Some limitations of this paper cannot be disregarded:
1.	Incomplete (flawed) interpretation: The authors' interpretations based on Figure 1 are questionable. Firstly, models trained on the Few-Class Regime (2-10 classes) have limited exposure to intra-class variability and may overfit to the specific characteristics of the few classes, potentially leading to inflated accuracies. Secondly, the small number (5) of test classes used in the experiments is highly susceptible to random fluctuations and statistical artifacts. The observed accuracies and the relative performance of different model sizes may be heavily influenced by these factors and not represent a genuine pattern or (increase or decrease) trend.
Moreover, the authors fail to provide any measures of uncertainty to support their claims. They do not report confidence intervals or error bars for the reported accuracies, making it difficult to assess the reliability and robustness of their findings. The observed differences in accuracies between larger and smaller models in the few-class regime may not be statistically significant, especially considering the limited number of test classes used in the experiments.
2.	Inadequate comparison with state-of-the-art: The paper does not provide a comprehensive comparison of the proposed Few-Class Arena (FCA) benchmark and Silhouette-based similarity score (SimSS) with existing state-of-the-art methods. The authors fail to justify the superiority of their approach over recently proposed few-shot learning algorithms, meta-learning techniques, and domain adaptation methods. Without a rigorous comparative analysis, the novelty and practical utility of the proposed solutions remain questionable.

**Opportunities For Improvement:**

1. Comparison with state-of-the-art: Include comprehensive comparative analysis of FCA benchmark and SimSS with relevant SOTA approaches, such as few-shot learning, meta-learning, and domain adaptation methods.
2. Handel overfitting and improve statistical quantification: Report confidence intervals and uncertainty estimation, discuss and mitigate potential overfitting in Few-Class Regime and further validate observed trends with additional experiments.

**Relation To Prior Work:**

Not clear. The authors do highlight the novelty of their work in proposing the first benchmark, Few-Class Arena (FCA), dedicated to the Few-Class Regime. They also emphasize the lack of comprehensive benchmarks for few-class research in the current literature. However, the illustration and discussion on how FCA differs from and improves upon existing benchmarks and difficulty measurement methods could be more explicit and detailed.

**Summary And Contributions:**

The paper introduces Few-Class Arena (FCA), a benchmark tool for evaluating efficient image classification models in the few-class regime, along with a scalable data loading approach and a novel similarity measure (SimSS). However, some interpretations are questionable due to potential overfitting and limited intra-class variability, which are susceptible to random fluctuations and statistical artifacts. The lack of uncertainty measures, comprehensive comparisons with SOTA methods, and ablation studies further limits the reliability and novelty of the findings.

This paper proposes Few-Class Arena (FCA), a unified benchmark focused on testing efficient image classification models for scenarios with a small number of classes (2-10). Main contributions are:
1.	Develops the Few-Class Arena (FCA) benchmark tool to facilitate research, analysis, and discovery for the few-class regime.
2.	Introduces a scalable few-class data loading approach to efficiently load images and labels from full datasets.
3.	Incorporates a novel Silhouette-based dataset similarity score (SimSS) as an inverse difficulty measure in FCA, showing high correlation with model performance.

---

> ### Author Rebuttal · Authors · 2024-08-16
>
> We thank reviewers for their constructive feedback on improving our work.
>
> __R3.1: "...the lack of ... comparisons with SOTA...", "Comparison with state-of-the-art: ... comparative analysis of FCA benchmark and SimSS with relevant SOTA approaches, such as few-shot learning, meta-learning, and domain adaptation ..."__ We address the concern regarding State-of-the-Art (SOTA) from the following aspects: (1) our proposed Few-Class Arena is the first benchmark tool that aims to facilitate scientific research, analysis, and discovery in the Few-Class Regime (Line 61-63). We have searched the literature on our best efforts but did not found such a comprehensive benchmark tool exists, meaning no other "SOTA" Few-Class benchmarks can be compared with; (2) we acknowledge that for a relatively well defined problem with several methods developed in the literature, a work with a new algorithm/model is supposed to be evaluated against those SOTA methods to quantify its advantages. However, our work, as a benchmark tool in this track, serves completely different purposes that we identified the neglected but less defined problems from many-class benchmarks to real-world applications (Line 27-32, 47-53), quantified the differences between full- and sub-models (Fig. 1, 3, 4).
>
> FCA benchmark serves as a tool to reveal insights in the Few-Class Regime, not intended for new method innovation, which is one reason why it is submitted in this Datasets and Benchmarks track instead of the regular one. In addition, Few-Class Regime is a new perspective and orthogonal to these methods (few-shot learning, meta-learning, and domain adaptation methods) which aims to improve a model's performance in a newly adapted environment.
>
> __R3.2: "...the absence of ... sensitivity analysis reduce the potential impact of the work.", "...Report confidence intervals and uncertainty estimation..."__ "variance" is commonly used as a means to quantify uncertainty, explicitly mentioned in Line 233, 234: "The variance, depicted by the light red areas, of the best models broaden dramatically for low $N_{CL}$, especially for $N_{CL}$ < 10." Variance is abundantly shown in the colored/gray areas in Fig. 1 (b) and all sub figures in Fig. 3-5 in the main draft and Fig. 3-22, 24 & 25 in the Appendix, totaling 25 figures. For analysis of variance (for uncertainty), please refer to our summarized key insights in 4.1 Results on FC-Full: "The variance, depicted by the light red areas, of the best models broaden dramatically for low $N_{CL}$, especially for $N_{CL}$< 10." in Line 234, 4.2 Results on FC-Sub: "variance reduces" for sub-models in Line 245.
>
> __R3.3: "...how the authors choose which few classes to choose to evaluate model performance."__ The procedure is documented in Sec. 3.2 Few-Class Dataset Preparation. Note that the whole benchmark serves as a generic tool to conduct experiments in the Few-Class Regime. Therefore the specific (few) classes are randomly selected from the full dataset to reduce bias. A seed number is documented for each subset to ensure reproducibility. While this procedure is described in the main draft, we use an example of a 1000 classes dataset to iterate again for clarity purposes: we can specify the list of number of classes ($N_{CL}$), say {4, 5}, and the list of the number of seeds ($N_{S}$), say {0, 1, 2}. For each $N_{CL}$, 4 for instance, we randomly extract 3 subsets all with 4 classes from the 1000 classes using seed {0, 1, 2}, respectively. We do the same thing for $N_{CL}=5$ to finish the list. In total we have 6 subsets. For each subset, we follow the conventional training and testing split and procedure.
>
> The classes in deployment depends on its specific requirement.
>
> __R3.4: "... why do we need a few-class arena and a systematic way to define which classes."__ The motivations are stated in the draft: "The de-facto benchmarks for valuating efficient vision models are large scale with many classes (Line 21)", "Real-world applications, however, typically comprise only a few number of classes (e.g, less than 10) [21, 22, 23] which we termed Few-Class Regime. To deploy a vision model pre-trained on large datasets in a specific environment, it requires the re-evaluation of published models or even retraining to find an optimal model in an expensive architectural search space [24]." (Line 24-30). "The lack of comprehensive benchmarks for few-class research impedes both researchers and practitioners from quickly finding models that are the most efficient for their dataset size. " (Line 50-52).
>
> __R3.5: "Model generalization is an important ... Evaluation on many classes is an advantage which can evaluate the generalizability to a broader data distribution... with fine-tuning ..., a well-trained and generalizable model can quickly adapt to specific tasks..."__ While we acknowledge the importance of the common belief that model generalization is an important character of model performance which is the aspect that most of the well developed many-class benchmarks are evaluating. We emphasize that we discover new problems in the less explored few-class regime -- a unique perspective in the literature. Since the objectives are different from different aspect: (1) general purposes for many-class benchmarks vs (2) efficient model selection for a practitioner, scientific discovery for our Few-Class Arena, the characteristics of a model will be different. The generalization-oriented design of many-class benchmarks can evaluate a model's generalization, but this ability does not matter in the few-class regime for practitioners. A practitioner mainly cares about a model's efficiency -- whether its accuracy is high enough with smaller model size and less compute cost. In summary, model generalization may not be an appropriate aspect in the few-class regime that we do not include the evaluation.
>
> For fine-tuning please refer to the response to R2.3.
>
> For ablation study please refer to the response to R1.4.

---

> > ### Author Response · Authors · 2024-08-26
> > **Additional Response to # Subsets (Not # Test Classes)**
> >
> > __R3.6: "...the small number (5) of test classes used in the experiments is highly susceptible to random
> > fluctuations and statistical artifacts. The observed accuracies and the relative performance of different
> > model sizes may be heavily influenced by these factors and not represent a genuine pattern or (increase
> > or decrease) trend."__
> >
> > We thank the reviewer for raising this concern,
> > (1) We would like to clarify the number of (5) is the __number of subsets__ (Fig. 1 caption) for each $N_{CL}$ instead of "number of test classes" mentioned in this concern (The subset sampling procedure is described in Sec. 3.2 Few-Class Dataset Preparation and response to R3.3). The number of test classes is determined by $N_{CL}$, __spreading__ from __few-class__ to __many-class__ regimes, specifically 2, 4, 10, 100, 200, 400, 600, 800 and 1000 in Fig. 1 (not just 5).
> >
> > (2) If the reviewer was attempting to ask about the random fluctuations and statistical artifacts due to the small number of subsets (5), we have added more experiment results with a ResNet small (ResNet18) and large (ResNet101) models on 5 more subsets (seed number 5-9) in addition to the 5 subsets in the original submission in the Table below. The newly integrated results are in the last two columns:
> >
> > |Model|$N_{CL}$|Full Top-1 Mean (5) | Full Top-1 Stdev (5) | Full Top-1 Mean (10)* | Full Top-1 Stdev (10)* |
> > |---|---|---|---|---|---|
> > |ResNet18|2|62.80|14.18|66.75|11.50|
> > |ResNet18|10|66.68|4.37|71.10|5.72|
> > |ResNet18|100|69.88|0.92|70.27|1.06|
> > |ResNet18|200|70.35|0.45|70.28|0.46|
> > |ResNet18|1000|69.90|-|69.90|-|
> > |ResNet101|2|72.20|13.31|76.70|10.60|
> > |ResNet101|10|75.48|2.97|79.30|5.07|
> > |ResNet101|100|77.99|1.36|78.41|1.21|
> > |ResNet101|200|78.41|0.31|78.40|0.26|
> > |ResNet101|1000|77.97|-|77.97|-|
> >
> >
> > |Model|$N_{CL}$|Sub Top-1 Mean (5) | Sub Top-1 Stdev (5) | Sub Top-1 Mean (10)* | Sub Top-1 Stdev (10)* |
> > |---|---|---|---|---|---|
> > |ResNet18|2|96.80|1.30|97.50|1.27|
> > |ResNet18|10|91.88|1.64|92.84|1.84|
> > |ResNet18|100|84.77|0.79|84.93|0.73|
> > |ResNet18|200|81.90|0.35|81.96|0.64|
> > |ResNet18|1000|69.90|-|69.90|-|
> > |ResNet101|2|95.80|1.30|96.80|2.10|
> > |ResNet101|10|90.72|2.40|91.92|2.39|
> > |ResNet101|100|87.48|0.52|87.51|0.47|
> > |ResNet101|200|85.32|0.49|85.55|0.67|
> > |ResNet101|1000|77.97|-|77.97|-|
> >
> > where ($N_{s}$) represents the number of subsets. *: Newly integrated results during the discussion period. Despite some small jitters in the full model results, which is commonly observed when dealing with large complicated datasets, sub-model results (10) in the last two columns are very close to the original ones (5) in the left two ones. Overall, these results are still __consistent__ with the summarized insights of Fig. 1 in the original submission. We use Fig. 1 as a motivation figure to illustrate the neglected behaviors in the community.
> >
> > We would like to emphasize the challenges of computation time of getting these experimental results especially during the discussion phase.
> >
> > We note that while these experiments can be done in the original dataset format, our proposed benchmark tool Few-Class Arena (FCA) scales the experiments of benchmarking models by a careful design of __configuration files__, __avoiding the need to duplicate data points__ for every additional few-class subset (Contribution 2 Line 64 and 3.1 Goals from Line 113 to 123).
> >
> > Also, we emphasize again that this work focuses on the development of the new tool (Few-Class Arena) while abundant analyses and results are presented to bring more attention to the research in the Few-Class Regime (neglected in prior work) with both challenges and opportunities. This work, submitted to the Datasets and Benchmarks Track, is designed to provide __comprehensive evaluation__ of algorithms/models/methods and therefore the innovation of these methods, such as few-shot learning, meta-learning, and domain adaptation methods etc. are great future work but beyond our scope.

---

> ### Author Response · Authors · 2024-08-31
> **Few-Class vs Fine-Tuning**
>
> __R3.7: "Evaluation on many classes is an advantage which can evaluate the generalizability to a broader data distribution. Even if the performance is originally not adapted to specific classes, with fine-tuning (many ways of training), a well-trained and generalizable model can quickly adapt to specific tasks. Thus the significance of this work is limited."__
>
> We thank the reviewer for the insightful and inspiring discussions regarding fine-tuning. To demonstrate the advantages of a few-class model (referred to as sub-model in our paper) over a fine-tuned model, we further conduct experiments inspecting a model's __sparsity__ -- the proportion of weights with zeros or near zero values in a neural network (NN). Our experiment design is inspired by the discover from prior works that a __sparser__ convolutional neural network (CNN) can lead to __speedup__ (e.g. acceleration in the FPGA level [1] or speedup 5.4x at 95% sparsity [2]), __reduction of memory footprint__ [3] and __benefit pruning__ [4].
>
> Specifically, we compare the __sparsity__ of a model with the same architecture, but the weights are trained separately in three different scenarios -- (1) full model: trained on __many classes__ on the full dataset; (2) ft-model: a model loaded (1)'s weights and __fine-tuned__ on a few-class dataset (a subset from the full dataset in (1)); (3) sub-model: trained on the same  __few-class dataset__ with (2) from scratch.
>
> We conduct experiments using 2 scales of ResNet (ResNet18 and ResNet50) on the CIFAR100 dataset. Specifically, a model in (1) or (3) is trained for 100 epochs. Following the standard fine-tuning process, a model in (2) first loads the weights trained in (1) and then is fine-tuned on the target few-class dataset for another 20 epochs. Results are shown in the two tables below. Each row represents 5 experiments. For (2) and (3), one row summarizes the results of 5 models on 5 few-class subsets sampled from the full class with seed numbers 0-4. We report the mean and standard deviation (STDDEV) of a model's Top-1 Accuracy (Top-1) and Sparsity. To understand how different scenarios (1)-(3) affect a model's effectiveness of learning visual features for efficiency, we compute _L1-norm_ of a convolutional filter's weights (3x3) to represents its importance [4]. Since a CNN's weights typically do not have exact zeros, we use a small threshold $\tau$ (default to 0.01) to determine zero values: if $||W_{ij}||<\tau$, we consider the weights of the filter $j$ in $i$th layer is $0$. Formally, the sparsity of a CNN is defined as $Sparsity=\frac{N_{0}}{N_{F}}$, where $N_{0}$ represents the number of zeros for filters while $N_{F}$ is the total number of filters.
>
> | Row # | Model | Scenario | $N_{CL}$ | Top-1 Mean (%) ↑ | Top-1 STDDEV | Sparsity Mean (%) ↑ | Sparsity STDDEV |
> |---|---|---|---|---|---|---|---|
> | 1 | ResNet18 | (1) full model | 100 | 76.11 | 0.37 | 94.41 | 0.20 |
> | 2 | ResNet18 | (2) ft-model | 2 | 87.90 | 1.71 | 94.32 | 0.19 |
> | 3 | ResNet18 | __(3) sub-model__ | 2 | __96.30 (+8.40)__ | 2.11 | __98.48 (+4.16)__ | 0.0063 |
> | 4 | ResNet18 | (2) ft-model | 4 | 87.60 | 6.92 | 94.32 | 0.19 |
> | 5 | ResNet18 | __(3) sub-model__ | 4 | __90.65 (+3.05)__ | 6.67 | __98.41 (+4.09)__ | 0.020 |
> | 6 | ResNet50 | (1) full model | 100 | 73.71 | 0.21 | 35.00 | 0.52 |
> | 7 | ResNet50 | (2) ft-model | 2 | 93.70 | 2.49 | 34.86 | 0.65 |
> | 8 | ResNet50 | __(3) sub-model__ | 2 | __95.10 (+1.40)__ | 1.92 | __82.06 (+47.20)__ | 0.28 |
> | 9 | ResNet50 | (2) ft-model | 4 | __90.55__ | 4.31 | 34.86 | 0.65 |
> | 10 | ResNet50 | __(3) sub-model__ | 4 | 88.25 (-2.30) | 7.72 | __80.30 (+45.44)__ | 0.32 |
>
> Overall, we verify that __(3) sub-models__ are __sparser__ while having competitive accuracies. We make two important observations: i) (3) sub-models are sparser than (1) full models & their corresponding (2) ft-models (Row 3 vs 2, Row 5 vs 4, Row 8 vs 7 and Row 10 vs 9); ii) although a (2) is fine-tuned on a few-class dataset, its sparsity still remains very close to its original (1) full model (Row 2 vs 1, Row 4 vs 1, Row 7 vs 6, Row 9 vs 6).
>
> We emphasize that in this under-explored few-class regime, (3) sub-models have the advantages of __higher sparsity__ that can lead to __higher efficiency__ compared to (2) ft-models. Neglected in existing many-class benchmarks, we unveil the discover to contribute to the literature with our proposed benchmark tool __Few-Class Arena__. We will incorporate these results in the camera-ready version, specifically by adding a subsection in Section 4 Experimental Results.
>
> [1] Lu et al. An efficient hardware accelerator for sparse convolutional neural networks on FPGAs. FCCM 2019.
>
> [2] Wang et al. Sparsert: Accelerating unstructured sparsity on gpus for deep learning inference. PACT 2020.
>
> [3] Hoefler et al. Sparsity in deep learning: Pruning and growth for efficient inference and training in neural networks. JMLR 2021.
>
> [4] Li et al. Pruning Filters for Efficient ConvNets. ICLR 2022.

---

### Official Review · Reviewer_C8Y1 · 2024-07-22

**Rating:** 5
**Confidence:** 3
**Clarity:** Yes

**Review:**

The target application of this benchmark is quite interesting, as it targets few-class image classification and appears to be the first benchmark for this application. However, my main concerns are about the evaluation fairness between many-class models and few-class models. Specifically:

(1) When using pre-trained full models for fewer classes, how is accuracy computed? For example, for a data sample of 5 classes (e.g., A, B, C, D, and E) with ground truth label A, if the 1000-class model gives the highest probability to one of the 5 classes but class A has the highest probability, is the 1000-class model considered correct in this case? If the proposed benchmark considers the 1000-class model incorrect in this case, then the results may be unfair for many-class models because the sub-models are trained and tested on few classes. Thus, even if the two types of models learn the same representation, they will have different accuracy, and the results will favor few-class models.

(2) How do you justify that real applications of few-class image classification need to specifically select model architectures instead of fine-tuning a smaller many-class model from various pre-trained models for few-class applications?

**Strengths:**

1. Less explored application: few-class image classification.

2. Includes both CNNs and ViTs.

3. High-quality figures and tables.

**Additional Feedback:**

None

**Correctness:**

The correctness of the comparison between many-class and few-class models needs more clarification, as mentioned in the review above.

**Documentation:**

Yes

**Ethics:**

No ethical concerns

**Limitations:**

The authors have discussed the limitations of this work. I agree that the coupled training procedure in the benchmark may affect the effectiveness of the insights derived from it.

**Opportunities For Improvement:**

It would strengthen the paper if the authors could address the two points of concern detailed in the review.

**Relation To Prior Work:**

Yes

**Summary And Contributions:**

This paper presents a benchmark for few-class image classification models. In particular, the authors include multiple CNNs and Transformer models in the benchmark and propose a Silhouette-based similarity score as a proxy for few-class dataset difficulty.

---

> ### Author Rebuttal · Authors · 2024-08-16
>
> We thank reviewers for the constructive comments.
>
> __R2.1: "When using pre-trained full models for fewer classes, how is accuracy computed? For example, for a data sample of 5 classes (e.g., A, B, C, D, and E) with ground truth label A, if the 1000-class model gives the highest probability to one of the 5 classes but class A has the highest probability, is the 1000-class model considered correct in this case?"__ Top-1 accuracy is computed. To be more specific, a 1000-class pretrained model make a prediction with its highest confidence, it computes the rate of the number of samples when the model's prediction matches the ground truth label over all test samples. When comparing a 1000-class model with a few-class one, the test set, including test classes and image samples, is always exactly the same for fairness.
>
> In the 5 classes example, since the class with the highest probability from the 1000-class is class A, its prediction is A which matches the ground truth A, this prediction is considered as correct.
>
> My understanding of this confusion is that (correct me if I am wrong) whether the small number of few classes (5 in this case) can have an effect on a larger Top-1 accuracy compared to 1000. In our benchmark, the answer is no.
>
> __R2.2: "If the proposed benchmark considers the 1000-class model incorrect in this case, then the results may be unfair for many-class models because the sub-models are trained and tested on few classes. Thus, even if the two types of models learn the same representation, they will have different accuracy, and the results will favor few-class models."__ For fair comparison, we ensure each Top-1 accuracy is reported on exactly the same test sets with few classes when comparing different models. With such a testing protocol, the difference between a pretrained many-class (named full-model in the paper) and a sub-model are mainly about the effectiveness of visual features learned in the deployed scenes. We argue that a full-model needs to learn features for all classes in order to achieve a high score in many-class benchmark, while the same amount of parameters (e.g. convolutional filters in CNN or attention maps in ViT-like models) can instead learn more discriminative features for few-class scenarios in Line 248 - 251 in the main manuscript. This seemingly obvious theory has been neglected as many-class benchmarks dominate most research communities.
>
> If two types of models, specifically full-model and sub-model, learn exactly the same representation and the same activation, these two models in theory will have exactly the same Top-1 accuracy score.
>
> We highlight in the paper (Line 27-30) that downloading the many-class pretrained models (especially those on ImageNet1K) and deploying it in an application is a common practice. But the aforementioned disadvantage is often neglected in many-class benchmarks (Line 235 - 238), which drives us to develop our proposed Few-Class Arena.
>
> __R2.3: "How do you justify that real applications of few-class image classification need to specifically select model architectures instead of fine-tuning a smaller many-class model from various pre-trained models for few-class applications?"__ We would like to clarify that we mentioned "model selection" in general in the paper, including selecting a model in various scales (such as EfficientNet-B0 to B7) and architecture types (CNNs v.s. ViTs), but rather than just from the model architecture dimension.
>
> In real-world scenario applications, there are accuracy requirements and hardware constraints under which the most efficient model is desired, in other words, a model with higher accuracy, smallest model size and/or lowest compute cost is preferred. The goal of model selection is to find the most efficient model that meets these requirements while real applications often have few classes for inference ([21, 22, 23] in Line 28).
>
> We note that our work focuses on the converged models in the few-class regime (Line 48). The main difference between a full-model and sub-model lies in the representations - whether they are for general purposes but may be extraneous for a full-model, or specific and good enough for the application with fewer parameters. Regarding fine-tuning, there are mainly two types of fine-tuning - (1) adjusting the whole model weights including the visual extraction backbone and the fully-connected layers and (2) freezing the backbone but only adjusting the fully-connected layers. From the representation learning's perspective, the former (1) (even for smaller many-class models from pretrained models) may have similar effects to models trained from scratch. But the latter (2) may still consist of extraneous parameters that might not be needed in the application but cannot be adjusted (due to freezing), hindering the potential efficiency that the model can achieve.
>
> Last but not least, we would like to emphasize that our proposed Few-Class Arena is not designed to provide an ultimate selection scheme but a platform that facilitates model selection for practitioners and for further scientific discovery (Line 61).

---

> > ### Author Response · Authors · 2024-09-01
> > **Few-Class v.s. Fine-Tuning**
> >
> > __R2.3.2: "How do you justify that real applications of few-class image classification need to specifically select model architectures instead of fine-tuning a smaller many-class model from various pre-trained models for few-class applications?"__
> >
> > We verify that a few-class model is __sparser__ than its corresponding many-class pre-trained model fine-tuned on few-class datasets. Please refer to the detailed response to R3.7.

---

### Official Review · Reviewer_knHF · 2024-07-25
**Few-Class Arena**

**Rating:** 5
**Confidence:** 3
**Clarity:** The paper is well written.

**Review:**

Quality and Clarity: The dataset and proposed metric are clearly stated, but the paper could benefit from additional tables or pipeline figures to enhance understanding.

Pros: The paper aims to design a more practical scenario with few classes, providing valuable insights for such applications.
Cons:
1. It is unclear how the method can be scaled up to other types of few-class scenarios since most experiments focus on ImageNet1K image classification tasks.
2. There is a lack of guidance on best practices for different scenarios, which could help in applying the method more broadly.

**Strengths:**

1. This work is the first to explore problems in the Few-Class Regime and introduces the benchmark tool Few-Class Arena (FCA) with three scenarios FC-Full, FC-Sub, and FC-Sim.
2. It includes a novel Silhouette-based similarity score, SimSS, which shows a high correlation (Pearson coefficient ≥ 0.88) with ResNet performance, leveraging CLIP and DINOv2.
3. Extensive experiments with ten models on ten datasets (2-1000 classes) on ImageNet, totaling 1591 training and testing runs, reveal new insights in the Few-Class Regime.

**Additional Feedback:**

Please refer to "Opportunities For Improvement".

**Correctness:**

The evaluation methods and experiment design are appropriate and performed correctly

**Documentation:**

The documentation provides sufficient detail to support reproducibility.

**Ethics:**

There are no or only very minor ethics concerns.

**Limitations:**

The authors have addressed some potential limitations and please refer to Opportunities For Improvement and Cons.

**Opportunities For Improvement:**

1. To improve clarity and comprehensiveness, it would be helpful to include visual aids such as tables and pipeline diagrams. For instance, a table comparing the differences among the four scenarios—Many-Class, FC-Full, FC-Sub, and FC-Sim—in terms of data involved and estimated training cost could be beneficial. Similarly, an overall figure illustrating the entire pipeline and benchmark tools would enhance understanding.

2. What is the ablation study performance for the proposed SimSS compared to the other three scores mentioned?

3. Additionally, expanding the scope of experiments to cover various few-class scenarios beyond ImageNet1K and providing practical guidelines for different use cases would enhance the utility and applicability of the method.

**Relation To Prior Work:**

It has clearly discussed difference from related works.

**Summary And Contributions:**

Numerous benchmark datasets with many classes (80-1000) support the evolution of Computer Vision architectures. While many models are tested on these datasets, real-world applications often involve fewer classes (2-10), making it hard to predict performance. Few-Class Arena (FCA) is a unified benchmark designed to test efficient image classification models for few classes. The ResNet family is evaluated on ImageNet subsets (2 to 1000 classes), and various CNN and Transformer architectures are tested across ten datasets using FCA. Additionally, FCA includes a difficulty measure based on class similarity to aid in model selection. FCA supports efficient machine learning in the Few-Class Regime, facilitating tasks like similarity proposals, lightweight model design, and scaling law discovery. It is user-friendly and easily extendable to new models and datasets, promoting future research.

---

> ### Author Rebuttal · Authors · 2024-08-16
>
> We thank reviewers for their comments and the opportunity to address their concerns.
>
> __R1.1: "It is unclear how the method can be scaled up to other types of few-class scenarios since most
> experiments focus on ImageNet1K image classification tasks."__ As a reminder, what we propose is a benchmark tool to reveal the problems and analyses yield insights in the novel few-class regime, instead of a method to improve an algorithm. This is the reason why this work is submitted to this track. We design our benchmark tool to be generic, listed in the 3.1 Goals section in Line 114 in the main draft. We randomly sample few-class subsets to represent different types of few-class scenarios, After conducting large-scale experiments, we analyze and study different models behaviors in this few-class regime.
>
> In total, ten datasets are included in the current benchmark (users can also easily extend to more datasets using our tool), on which FC-Full, FC-Sub, FC-Sim experiments are conducted. These datasets represent diverse real-world scenarios, including CalTech101, CalTech256, CIFAR100, ImageNet1K for common objects, CUB200 for birds, Food101 for food, GTSRB43 for traffic signs, Indoor67 for indoor scenes, Quickdraw345 for sketch drawings, Textures47 for textures. Each experiment represents one specific few-class scenario where classes are sampled from the original full dataset with a specific number of classes $N_{CL}$ and seed number $N_{S}$ (for reproducing results), described in Line 128. ImageNet1K is generally well accepted as a general object dataset, where we conduct experiments in a deeper level than others to further support our findings. For all datasets, we show the results in Fig. 3 - 5 in the main manuscript and Fig. 3 - 22 and Fig. 24 & 25 in the Appendix. We have included 5 ($N_{CL}$) * 5 (# subsets) * 10 (# datasets) = 250 different few-class scenarios spanning ten diverse datasets. In contrast, most of the current algorithm papers report performance typically in one scenario per dataset unless otherwise specified.
>
> __R1.2: "There is a lack of guidance on best practices for different scenarios, which could help in applying the
> method more broadly."__ Thanks for mentioning the guidance as this is actually the main reason why we incorporate a difficulty measure as a function of class similarity (Line 13) and design the Few-Class Similarity Benchmark (FC-Sim). The similarity score (SimSS) in Line 196, acts as a guidance of measuring a dataset's difficulty in its specific scenario. Different scenarios present various SimSS scores shown in Fig. 23 in the Appendix. This score, which is highly correlated with DCN-Sub (the best model trained in each scenario) as shown in Fig. 5 (c) (d) in the main manuscript, gives guidance to practitioners in estimating the best model's performance for selection.
>
> We would like to mention again that what we claim as the main contributions are the creation of this Few-Class Arena benchmark tool and the key insights from the extensive experiments.
>
> __R1.3: "To improve clarity and comprehensiveness, it would be helpful to include visual aids such as tables and pipeline diagrams. For instance, a table comparing the differences among the four scenarios—Many-Class, FC-Full, FC-Sub, and FC-Sim—in terms of data involved and estimated training cost could be beneficial. Similarly, an overall figure illustrating the entire pipeline and benchmark tools would enhance understanding."__ Fig. 1 serves as a summary of findings that involves Few-Class, Many-Class, FC-Full and FC-Sub. FC-Full and FC-Sub results are shown jointly in Fig. 4 in the main draft and Fig. 13 - 22 in the Appendix. Training cost is mentioned in A.9 Experiments Compute Resources (Line 478) in Appendix. We will include an overall figure of the entire pipeline in the camera-ready version.
>
> __R1.4: "What is the ablation study performance for the proposed SimSS compared to the other three scores mentioned?"__ We thank reviewers for the feedback. We include the ablation study results below. Specifically, we follow the same evaluation protocol to SimSS in Sec. 4.3 Results on FC-Sim, where we compute the correlation between a similarity score and DCN-Sub
>  (the highest empirical accuracy a model can achieve in a subset). We would like to clarify although we mention three other scores apart from our proposed SimSS, SimSS (Equation 4) actually consists of only two scores (1) Intra-Class Similarity $S_{\alpha}$ and (2) Nearest Inter-Class Similarity $S_{\beta}'$ instead of three. We derive (2) from Inter-Class Similarity $S_{\beta}$. Therefore for ablation study, we conduct experiments on these two individual components, specifically the correlation between (a) DCN-Sub and $S_{\alpha}$ and (b) DCN-Sub and $S_{\beta}'$ measured by Pearson Correlation coefficients $r$ with a p-value. We conduct experiments using both CLIP and DINOv2 as general visual backbone as similarity base function. The results are reported as follows:
>
> $r$(DCN-Sub, $S_{\alpha}[CLIP]$) = 0.21, p-value = 0.26
>
> $r$(DCN-Sub, $S_{\alpha}[DINOv2]$) = -0.046, p-value = 0.81
>
> $r$(DCN-Sub, $S'_{\beta}[CLIP]$) = -0.048, p-value = 0.80
>
> $r$(DCN-Sub, $S'_{\beta}[DINOv2]$) = -0.24, p-value = 0.19
>
> $r$(DCN-Sub, $SimSS[CLIP]$) = __0.90__, p-value = 8.8e-12
>
> $r$(DCN-Sub, $SimSS[DINOv2]$) = __0.88__, p-value = 2.4e-10
>
>
> where the first four rows show the results using only one similarity metric by ablating the other one (poorly correlated with DCN-Sub, $r$ < 0.25 and the last two rows of our proposed similarity score (extremely highly correlated with DCN-Sub, $r$ >= 0.88), demonstrating the effectiveness of our design choice. We will include these ablation study results in the camera-ready version.

---

> > ### Author Rebuttal · Authors · 2024-09-01
> >
> > __R1.3.2: "To improve clarity and comprehensiveness, it would be helpful to include visual aids such as tables and pipeline diagrams. For instance, a table comparing the differences among the four scenarios—Many-Class, FC-Full, FC-Sub, and FC-Sim—in terms of data involved and estimated training cost could be beneficial. Similarly, an overall figure illustrating the entire pipeline and benchmark tools would enhance understanding."__
> >
> > We thank the reviewer's constructive feedback and the opportunity to improve our work. We have added the overall figure to illustrate the pipeline in the attachment "overview.pdf" and the first figure in the public repository: https://github.com/fewclassarena/fca. We will incorporate this figure in the camera-ready version, specifically in Section 3 Few-Class Arena (FCA).

---

### Author Rebuttal · Authors · 2024-08-16

General rebuttal:
We thank reviewers for their positive comments on recognizing the contributions of this work, particularly in the following aspects:

__(1) Novelty__:
- [Reviewer knHF] "This work is the first to explore problems in the Few-Class Regime and introduces the benchmark tool Few-Class Arena (FCA) with three scenarios FC-Full, FC-Sub, and FC-Sim.", "clearly discussed difference from related works."
- [Reviewer C8Y1]"Less explored application: few-class image classification.",
- [Reviewer zZVd] "Systematically explore the few-class regime (2-10 classes), which is highly relevant to real-world applications but largely overlooked in previous research focused on many-class datasets (80-1000 classes)".

__(2) Presentation__:
- [Reviewer C8Y1] "High-quality figures and tables.".
- [Reviewer zZVd] "...is well written.", "well-structured and clearly written", "constructed in a sound way", "well-written and clear, with a concise abstract and clearly defined goals."

__(3) Extensive Evaluation__:
- [Reviewer knHF] "Extensive experiments with ten models on ten datasets (2-1000 classes) on ImageNet, totaling 1591 training and testing runs, reveal new insights in the Few-Class Regime.",
- [Reviewer C8Y1] "Includes both CNNs and ViTs.",
- [Reviewer zZVd] "systematic evaluation of a diverse range of models (CNNs and Transformers) on multiple datasets across a wide spectrum of class sizes", "spanning ten models and ten datasets, totaling 1,591 training and testing runs".

__(4) Documentation__:
- [Reviewer knHF] "The documentation provides sufficient detail to support reproducibility.",
- [Reviewer zZvD] "The authors provide a good amount of documentation on data collection and benchmark organization. The provided code repository, along with the details on dataset preparation and the use of the MMPreTrain framework, offers a solid foundation for reproducibility."

Below, we address the concerns raised by each reviewer.

---

### Decision · Program_Chairs · 2024-09-26

**Decision:**

Reject

**Comment:**

This paper received three reviews and all are negative. The main concern is this work just selected and constructed a benchmark from existing datasets. It fails to clarify how its approach can be extended to different types of few-class scenarios and provide guidance on best practices for practitioners to apply the method effectively in various contexts.